# New insights in the coordinated amidase and glucosaminidase activity of the major autolysin (Atl) in *Staphylococcus aureus*

Mulugeta Nega[1], Paula Maria Tribelli[1,2], Katharina Hipp [3], Mark Stahl[4] & Friedrich Götz [1✉]

After bacterial cell division, the daughter cells are still covalently interlinked by the peptidoglycan network which is resolved by specific hydrolases (autolysins) to release the daughter cells. In staphylococci, the major autolysin (Atl) with its two domain enzymes, N-acetylmuramyl-L-alanine amidase (AmiA) and β-N-acetylglucosaminidase (GlcA), resolves the peptidoglycan to release the daughter cells. Internal deletions in each of the enzyme domains revealed defined morphological alterations such as cell cluster formation in ΔamiA, ΔglcA and Δatl, and asymmetric cell division in the ΔglcA. A most important finding was that GlcA activity requires the prior removal of the stem peptide by AmiA for its activity thus the naked glycan strand is its substrate. Furthermore, GlcA is not an *endo*-β-N-acetylglucosaminidase but an *exo*-enzyme that cuts the glycan backbone to disaccharides independent of its *O*-acetylation modification. Our results shed new light into the sequential peptidoglycan hydrolysis by AmiA and GlcA during cell division in staphylococci.

[1] Microbial Genetics, Interfaculty Institute of Microbiology and Infection Medicine Tübingen (IMIT), University of Tübingen, Auf der Morgenstelle 28, D-72076 Tübingen, Germany. [2] Departamento de Química Biológica FCEyN-UBA, Buenos Aires, Argentina. [3] Max Planck Institute for Developmental Biology, Max-Planck-Ring 5, D-72076 Tübingen, Germany. [4] Zentrum für Molekularbiologie der Pflanzen (ZMBP), University of Tübingen, Auf der Morgenstelle 32, D-72076 Tübingen, Germany. ✉email: friedrich.goetz@uni-tuebingen.de

CW biosynthesis and degradation during bacterial cell division is a balanced process. The CW remodeling and also cell separation is carried out by bacterial hydrolases. To maintain cell integrity and to avoid cell lysis, each cell wall hydrolase has a specific target in PG processing: (i) the amidase cleaves the amide bond between N-acetylmuramic acid and the L-alanine residue of the stem peptide, (ii) glucosaminidase catalyzes the hydrolysis of the glycosidic linkages, whereas (iii) peptidase cleaves amide bonds between amino acids within the PG chain[1]. Most bacteria produce a whole range of different PG hydrolases that perform different tasks such as nicking of PG to incorporate new monomers, remodeling and turnover of PG during growth, cell division, and cell separation[2]. In staphylococci, the last step in cell division is the separation of the two interlinked daughter cells by cell wall (CW) hydrolases. The separation of daughter cells is mainly catalyzed by the so-called "major autolysin", Atl. This is a secreted bifunctional multidomain enzyme typical for the genera *Staphylococcus* and *Macrococcus*[3]. Atl of both *S. aureus* and *S. epidermidis* is processed into two bacteriolytic active domains, the ≈62 kDa N-acetylmuramyl-L-alanine amidase (AmiA) and the ≈51 kDa endo-beta-N-acetylglucosaminidase (GlcA)[4–6]. The 51 kDa GlcA from *S. aureus* was described as an endo-beta-N-acetylglucosaminidase[7]. Deletion of the entire *atl* gene in *S. epidermidis* led to huge cell cluster formation indicating that daughter cell separation was severely affected and the supernatants lacked five prominent proteins[4]. This defect could largely be complemented by *atl* or *amiA* alone. Interestingly, to compensate for the loss of *atl*, staphylococci induce another 35 kDa autolysin (*aaa*) with a cysteine, histidine-dependent amidohydrolase/peptidase (CHAP) domain[8,9]. Atl proteins mediate attachment of staphylococcal cells to polymer surfaces and enhance biofilm formation[4,10,11].

Atl is proteolytically processed to the two enzymes AmiA and GlcA[4,5,12]. This processing occurs in such a way that each of the enzymes is supplied with repeat domains generating AmiA-R1ab/R2ab and R3ab-GlcA (Fig. 1a). The repeat regions target AmiA

and GlcA to the septum in two ways: (a) the repeats are repelled by the wall teichoic acid (WTA), which is mainly present in the mature CW[13], and (b) the repeats bind to the lipoteichoic acid (LTA), which is localized in the septum[14]. In this way, the repeats direct AmiA and GclA to the septum, where they can optimally carry out the final step of cell division by resolving the PG-interlinked daughter cells.

To obtain more insight into the amidase, the structure of the soluble 214 aa catalytic domain of amidase from *S. epidermidis* (AmiE) and *S. aureus* (AmiA) were determined[15,16]. Both amidases adopt an almost identical globular fold, with several alpha-helices surrounding a central beta-sheet. In the active site, an essential zinc ion is buried which stabilizes the transition state during catalysis. The minimal substrates for AmiE and AmiA is the muramyltripeptide which must contain a di-amino acid lysine. AmiE and AmiA reveal a striking and unexpected homology to the family of the mammalian PG recognition proteins (PGRPs), some of which also possess amidase activity[16].

The Atl amidase is composed of the catalytic Ami domain and repeats R1ab and R2ab. These three modules are separated by two linkers, L1 and L2[14,17]. The two repeats (R1ab and R2ab) target the Ami to the septum where LTA serves as a receptor. The crystal structure of R2ab reveals that each repeat folds into two half-open beta-barrel; and small-angle X-ray scattering of the mature amidase reveals the presence of flexible linkers (L1 and L2) separating the Ami, R1ab, and R2ab units. The linkers act as a hinge region allowing high flexibility and fidelity of the amidase domain. All data suggest that the repeats direct the catalytic amidase domain to the septum, where it can optimally perform the final step of cell division[14]. The functional and structural analysis of the major autolysin (Atl) in *Staphylococcus* has been reviewed by Götz et al.[18].

Although several studies have focused on the functional and the structural analysis of Atl, little is known about the interplay of AmiA and GlcA during cell division and separation. Here, we compared morphological changes of *S. aureus atl* mutants carrying deletions of the *amiA* (Δ*amiA-R1-R2*), *glcA* (Δ*R3-glcA*) domains and the entire *atl* (Δ*atl*). In the past, AmiA activity has been investigated mainly with synthetic substrates or in zymograms, here we demonstrate its activity with its natural substrate PG. Moreover, we demonstrate that the glucosaminidase GlcA is not an endo- but an exo-beta-N-acetylglucosaminidase which is only active with the naked glycan strand after AmiA has cleaved off the crosspeptides. This implies that the cutting (resolution) of PG occurs in a defined order, in which AmiA first hydrolyzes the crosspeptides, and only then can GlcA dissect the glycan strand to disaccharides units.

## Results

### Morphological characterization of *S. aureus amiA*, *glcA,* and *atl* mutants by scanning and transmission electron microscopy (SEM and TEM)

The staphylococcal major autolysin, Atl, is a multidomain protein composed of the signal peptide (SP), the propeptide (PP), the catalytic domain (cat) of the amidase AmiA followed by the repeat domains R1 and R2, and the glucosaminidase domain (GlcA) which is preceded by the R3 repeat domain (Fig. 1a). To compare the morphological consequences of each enzyme, we constructed internal deletion mutants in the *S. aureus* SA113 strain. In SA113Δ*amiA*, the *amiA* domain together with the repeats R1 and R2 were deleted. In SA113Δ*glcA* the R3 and *glcA*, whereas in Δ*atl* the entire *atl* gene was deleted (Fig. 1b).

Comparative scanning electron microscopic (SEM) analysis of SA113 and the mutants Δ*amiA*, Δ*glcA* and Δ*atl* grew to mid-exponential phase revealed that in all mutants, the cell separation was affected as manifested by increased cell cluster formation, indicating that both, AmiA and GlcA contribute to cell separation

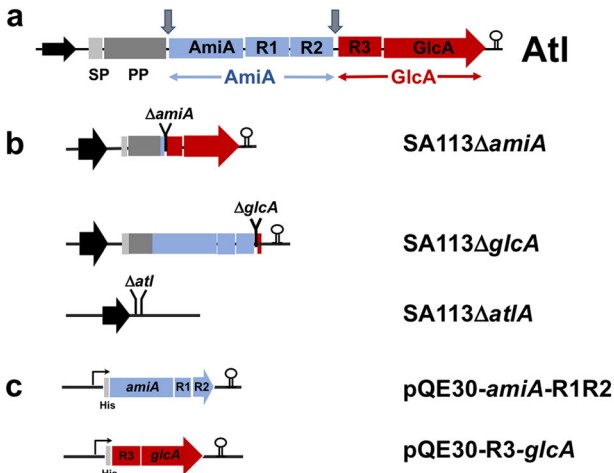

**Fig. 1 Organization of the domain structure of the *S. aureus* major autolysin (Atl) and construction of mutants and expression vectors. a** Atl is organized as a multidomain protein starting with the signal peptide (SP), which is followed by the propeptide (PP), the amidase (AmiA-R1R2), and the glucosaminidase (R3-GlcA) domains. A certain proportion of Atl is post-translocationally processed as indicated by arrows. The repeats, connected by a linker, represent LTA-binding domains that target the enzymes to the septum. **b** Deletion constructs of the *amiA* (Δ*amiA*) and *glcA* (Δ*glcA*) domains and the entire *atl* (Δ*atl*) gene. **c** *amiA* and *glcA* expression constructs in pQE30 fitted with an N-terminal His-tag.

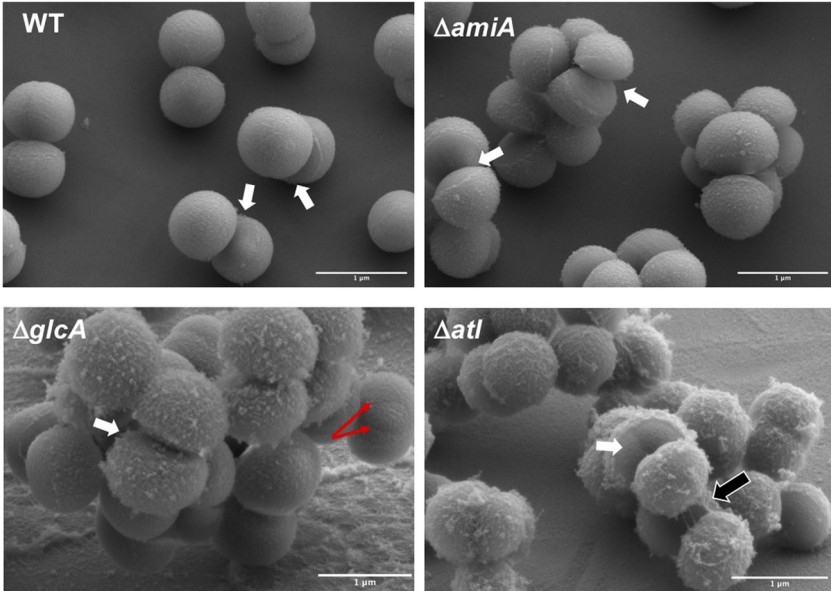

**Fig. 2 Comparative SEM images of exponential grown SA113 and its mutants.** WT cells have smooth old CW with well-separated daughter cells. Some cell pairs are already broken open by a mechanical crack popping apart the daughter cells (white arrows). In Δ*amiA*, cells are noticeably more clumped with the popped cells while the old CW is rougher than in the WT. In Δ*glcA*, the roughness of the old CW was further intensified with pockmarked elevations, and in some cells, asymmetric septa were formed (red arrows). Finally, Δ*atl* showed the highest clumped phenotype, with lots of separate cells that are still connected by thread-like PG structures (black arrow); the rough surface of the old CW is full with pockmarked elevations which represent accumulations of unprocessed PG. Popped cells (white arrows) are seen in all mutants; red arrows, indicate asymmetric septa; black arrow, indicates thread-like PG structures by which the neighboring cells are still connected. Scale bars 1 μm.

(Fig. 2). In both WT and the mutants, popped daughter cells were seen that were mechanically cracked apart as described by Zhou et al.[19]. Popped daughter cells were seen in all mutants indicating that this part of cell division is not affected in the mutants. In Δ*amiA*, cells are more clumped and the old CW is rougher than in the WT. In Δ*glcA* the roughness of the old CW was further intensified with pockmarked elevations and one can see that in some cells, asymmetric septa were formed (red arrows). Finally, Δ*atl* showed the highest clumping phenotype and most rough cell surface; there were lots of separate cells that are still connected by thread-like PG structures (Fig. 2). This difference in cell cluster formation in the mutants was further confirmed by forward scatter flow cytometric measurements of WT SA113, Δ*amiA*, Δ*glcA,* and Δ*atl* cells grown to mid-exponential phase. The results clearly show that the mean cluster size increases in the order WT < Δ*amiA* < Δ*glcA* < Δ*atl* confirming the previous microscopic observations (Supplementary Fig. 4).

In order to confirm our hypothesis that the thread-like structures are filaments of unprocessed "old" PG, we digested heat-inactivated mutant Δ*atl* cells that were grown to mid-exponential phase with purified AmiA and GlcA. A comparison of the digested and undigested samples with SEM analysis showed that the clumping phenotype of the undigested sample is completely abolished and cells were separated. More importantly, the rough surface becomes smooth confirming that the "old" PG was processed by the externally added enzymes (Fig. 3).

Through cross-sectional analysis of dividing cells using TEM, it has become more precisely visible which deformations occur during cell division (Fig. 4). In the WT strain, the cells are dividing into three alternating perpendicular planes, with sister cells remaining attached to each other after division and the resulting point of attachment was usually not exactly at the point corresponding to the center of the previous septal disk as described already by Tzagoloff and Novick 1977[20]. In Δ*amiA*, some thread-like structures were released, most likely sheared-off CW residues; otherwise the division planes looked normal like in

the WT. In Δ*glcA*, however, there was a high proportion of cells where the division plane was asymmetric giving rise to "kidney" like cell arrays. Apparently, the perpendicular sequence of the cell division plane was disturbed (Fig. 4 and Supplementary Fig. 1). In Δ*atl* such asymmetric septa formations were not seen; however, the clumping phenotype is extreme, and even separated cells are still connected by thread-like PG structures. The different morphological features of the mutants indicate that AmiA and GlcA are important determinants of cell division and separation with distinct functions.

**Deletion of *atl* results in decreased PG cross-linking**. For the analysis of PG cross-linking, PG isolated from the SA113, its Δ*atl* mutant and complemented mutant Δ*atl*(pRC14) were resuspended with buffer, adjusted to OD 3.0 and digested with mutanolysin. The released muropeptides were resolved with RP-HPLC, and area units of each fragment were compared relative to the area units of the whole chromatogram (Fig. 5a). The total composition of the PG fragments was largely the same in SA113 and its Δ*atl* mutant. However, in the Δ*atl* mutant, the short PG fragments (mono- to pentamer) were increased 1.5 to twofold, while the longer PG fragments (≥octamer) were about twofold decreased (Fig. 5a, b). The shift from polymeric to oligomeric PG fragments in the Δ*atl* mutant was also confirmed by the decrease of the hump at later elution times (>100 min). These results show that in the Δ*atl* mutant, not only cell separation but also PG cross-linking was impaired. To see which enzyme contributes most to PG cross-linking, we also analyzed Δ*amiA* and Δ*glcA* mutants. In both mutants, PG cross-linking was decreased as well but not as low as in the Δ*atl* mutant suggesting that both the AmiA and GlcA contribute to PG cross-linking (Supplementary Fig. 3).

**Digestion of PG by AmiA caused the release of cross-linked peptides of varying length**. As a control, we digested purified *S*.

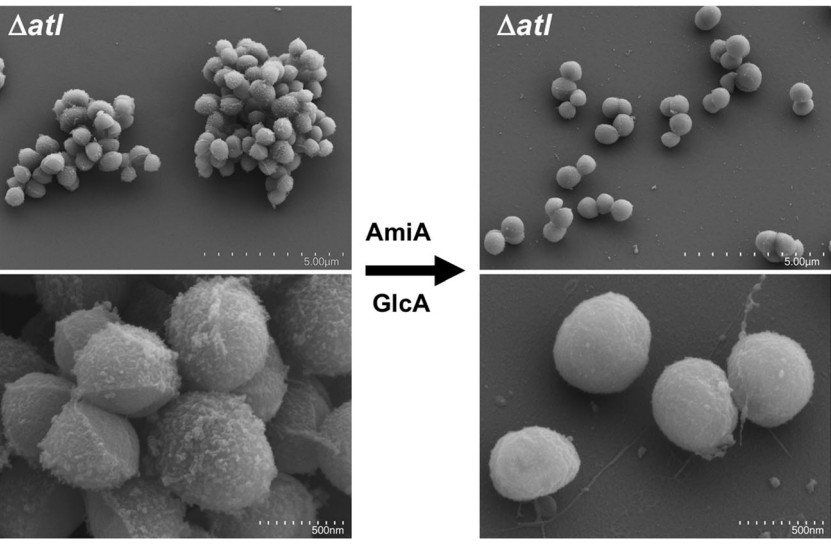

**Fig. 3 Scanning electron microscopic (SEM) analysis of enzyme digested SA113Δ*atl* cells.** Cells grown to mid-exponential phase were centrifuged and resuspended in 50 mM phosphate buffer pH 7.0 and heat-inactivated at 65 °C for 30 min. A portion of it was digested with purified AmiA and GlcA. SEM analysis of both the digested and undigested cells shows smoothening of the surface and removal of old PG from the surface. The images at the bottom are magnifications (scale bar 500 nm) of the same upper sample image (scale bar 5 μm) showing details of surface smoothness and separation.

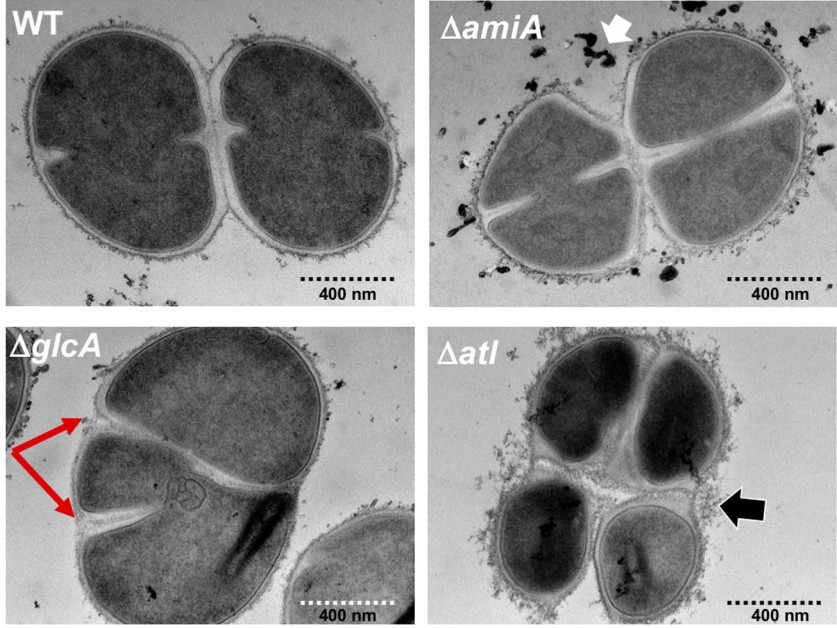

**Fig. 4 Transmission electron microscopic (TEM) images of exponential grown SA113 and its mutants.** Through cross-sectional analysis of dividing cells using TEM, it is more precisely visible which deformations occur during cell division. In Δ*amiA* cells, the old CW is rougher than in the WT but division symmetry is unaffected. In Δ*glcA*, the asymmetric septa formation is now clearly visible than in the SEM images (red arrows). In Δ*atl*, even separated cells are still connected by thread-like PG structures (black arrow). Scale bars 400 nm.

*aureus* 113 PG with the muramidase mutanolysin and separated the solubilized digestion products by RP-HPLC. As described previously[21], we obtained the typical peak pattern from monomeric up to heptameric PG fragments followed by a hump of the unresolved polymeric fraction (Fig. 6a). AmiA digestion was carried out in the same way as with mutanolysin. AmiA (AmiA-R1-R2) was isolated from *E. coli* M15 (pQE30Ω*amiA-R1-R2*) (Fig. 1c) and purified via its N-terminal His$_6$-tag. After the digestion of PG with AmiA, the solubilized PG fragments were analyzed by RP-HPLC. We could resolve between 15 and 20 peaks (Fig. 6b). Mass spectrometric (MS) analysis of the major peaks revealed that the amidase effectively cleaved between the N-

acetylmuramic acid and the L-Ala of the stem peptide, thus releasing cross-linked peptides of varying lengths, from mono- up to >12-mer (Fig. 7a and Supplementary Table 2). Each peak showed a mass value that corresponds to a tetrapeptide-pentaglycine fragment (peptide monomer) that is exactly the fragment of the next cross-linking (Fig. 7b).

**GlcA requires the prior removal of stem peptide by amidase for its activity.** Like AmiA, the glucosaminidase, GlcA, was isolated from *E. coli* M15 (pQE30Ω*R3-glcA*) (Fig. 1c) and purified via its N-terminal His$_6$-tag. While AmiA digested PG shows the typical

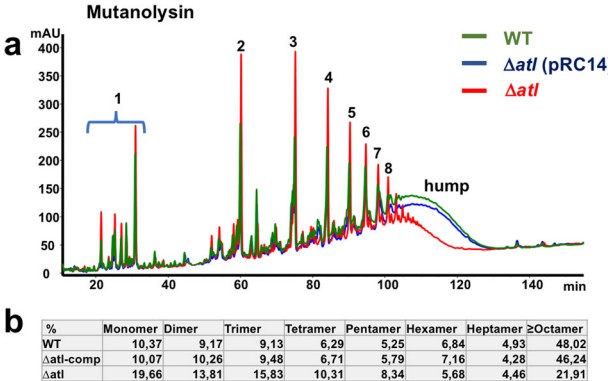

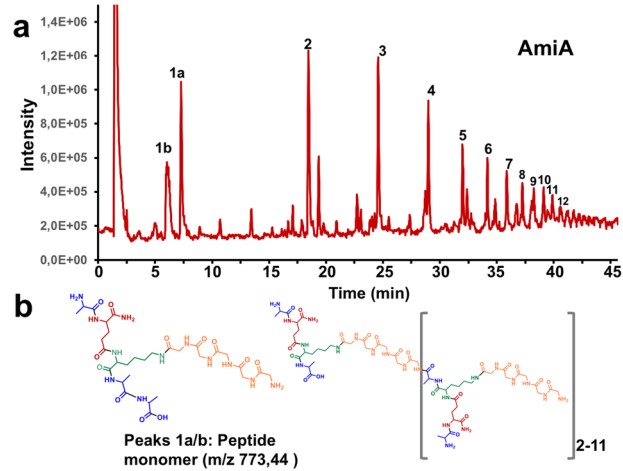

**Fig. 5 RP-HPLC profiles of solubilized muropeptides of mutanolysin digested PG. a** Muropeptide profile of WT, Δ*atl* and the plasmid complemented Δ*atl* (pRC14) strains. The peaks 1–8, up to a retention time (RT) of ∼100 min, represent soluble monomeric to octameric muropeptide fragments. The peak areas of the smaller fragments (peaks 1–5) is much higher in the Δ*atl* mutant than in the WT, indicating that in Δ*atl*, PG cross-linking is decreased. Higher-order cross-linked polymeric muropeptides (RT > 100 min) are characterized by a hump of the unresolved polymeric fraction. **b** Percent distribution of muropeptide fragments of the WT, plasmid complemented Δ*atl*, and Δ*atl* mutant. The figure is representative of three independent experiments.

**Fig. 7 HPLC/MS profile of AmiA solubilized staphylococcal PG. a** showing peaks (1–12) of the peptides released by AmiA hydrolysis of PG between MurNAc of the glycan backbone and L-Ala of the stem peptide in increasing length. The mass of each numbered peak corresponds to the number of crosspeptides composed of the stem peptide and the pentaglycine bridge (monomer). **b** Structural representation of the peptide peaks 1a/b (peptide monomer) and the extension of the monomeric structure up to its 12-mer shown in square brackets.

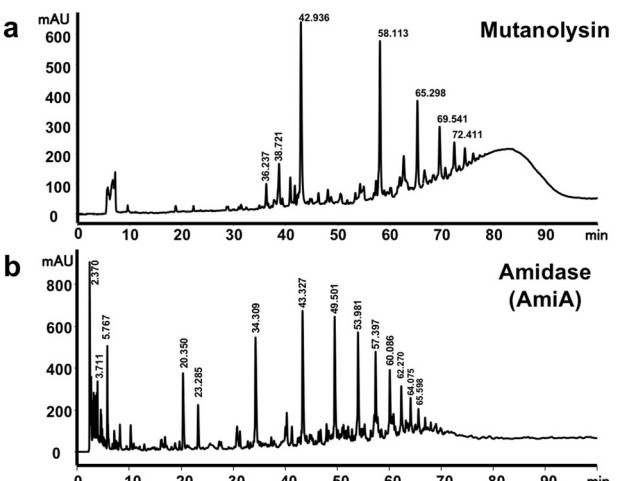

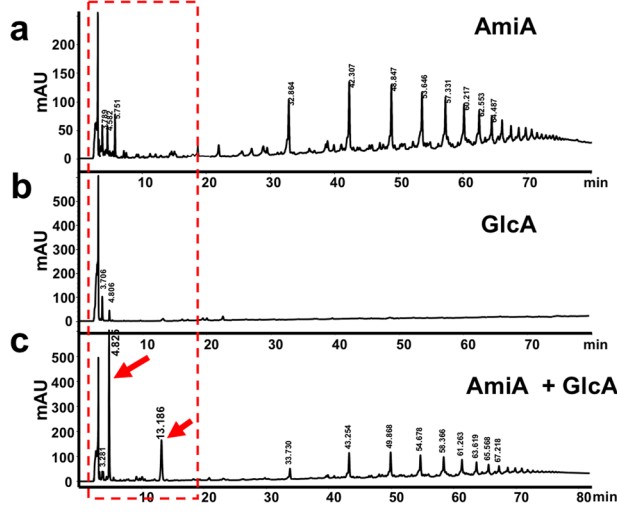

**Fig. 6 Muramidase (mutanolysin) and amidase (AmiA) digestion pattern of staphylococcal PG. a** RP-HPLC analysis of soluble PG of *S. aureus* SA113 released after digestion with mutanolysin showing the characteristic peak pattern of monomeric up to heptameric PG fragments ending in a hump of an unresolved polymeric fraction as described previously[21]. **b** HPLC analysis of soluble PG fragments released after digestion with AmiA shows a distinctly different peak pattern.

**Fig. 8 RP-HPLC profile of soluble fragments obtained after hydrolysis of PG*. a** with AmiA, **b** with GlcA, **c** with GlcA after prior digestion with AmiA; AmiA was heat-inactivated at 95 °C for 3 min before digestion with GlcA. After the double digestion, two distinct peaks at RT 4.8 and 13.1 min appear as main products, indicated by red arrows. *Please note the different scales in **a** = 250, **b** = 600, and **c** = 600 mAU. All samples are heat-inactivated at 95 °C for 3 min before analysis.

soluble PG fragments (Fig. 8a), no soluble PG fragments were observed with GlcA (Fig. 8b), suggesting that GlcA cannot cleave PG. Therefore, we assumed that GlcA is active only if PG is predigested with AmiA. To verify this assumption, we predigested PG with AmiA for 16 h, inactivated the AmiA by heating to 95 °C for five minutes, followed by digestion with GlcA for a further 16 h. Now two main peaks appeared at RT 4.8 and 13.1 min, while the peaks specific to AmiA remain unaltered (Fig. 8c). The masses of the peaks at RT 4.8 and 13.1 min were m/z 498 and m/z 540 (reduced form) (Fig. 9a, b), indicating that the peak at RT 4.8 corresponds to MurNac-GlcNac disaccharides and the one at RT 13.1 to O-acylated MurNac-GlcNac (Fig. 9c). This result shows

that GlcA is only active with an unsubstituted glycan chain that is free of peptide subunits. It also shows that O-acetylation of MurNAc does not affect GlcA activity.

**GlcA is not an endo- but an exo-β-N-acetylglucosaminidase.** Since GlcA treatment of the AmiA-predigested PG only resulted in disaccharide products, we assumed that GlcA is not an endo- but must be an exo-β-N-acetylglucosaminidase. If GlcA is an endoenzyme, as described in the literature[5,22], we should see glycan products of different sizes in the course of time. We, therefore, digested AmiA predigested PG with GlcA and followed the product formation over a period of 0 to 8 h. If GlcA is an

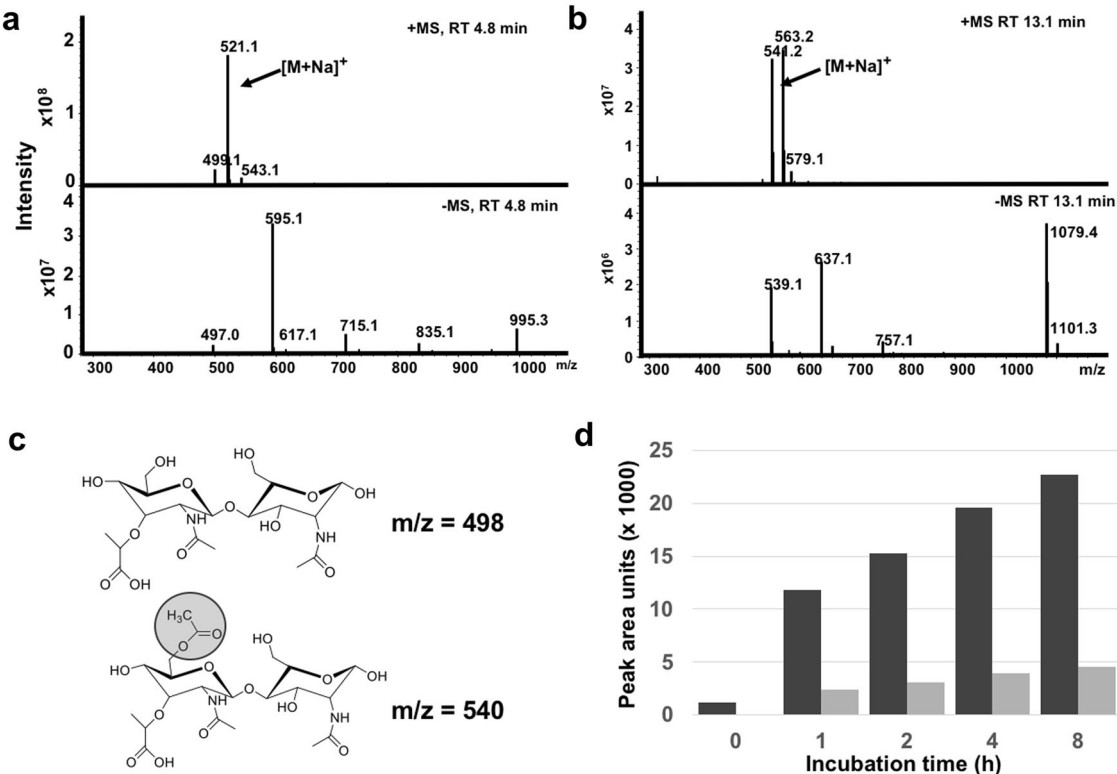

**Fig. 9 Mass spectrometric and time-lapse profile of the GlcA digestion disaccharide products, MurNac-GlcNac.** The mass of the two distinct HPLC peaks at Rt 4.8 min (**a**) and 13.1 min (**b**) shown in Fig. 7c was determined. **c** The mass of m/z = 498 corresponds to MurNAc-GlcNAc and of m/z = 540 to O-acetylated MurNAc-GlcNAc. **d** Time course of MurNAc-GlcNAc release by AmiA and GlcA digestion measured by RP-HPLC. The black bar represents MurNAc-GlcNAc and the gray bar O-acetylated MurNAc-GlcNAc. The figure is representative of three independent experiments.

exoenzyme, we should only see disaccharide products with increasing concentrations over time. Indeed, HPLC analysis of the samples at different time points showed that only MurNAc-GlcNAc disaccharide peaks appeared that increased with time (Fig. 9d). This result clearly shows that first, the substrate of GlcA is the naked glycan strand devoid of cross-linking peptides, and second, GlcA is an exo-β-N-acetylglucosaminidase that acts on the outer tip of the glycan chain releasing only MurNAc-GlcNAc-disaccharides.

## Discussion

In staphylococci, separation of the daughter cells is mainly catalyzed by the so-called "major autolysin", Atl. This bifunctional multidomain protein consists of the two murein hydrolase domains, the amidase (AmiA) and glucosaminidase (GlcA) with their adjoining repeats. The cell morphological changes of mutants in which the entire *atl* gene is deleted have been described in *S. aureus* and *S. epidermidis*[4,23]. The *atl* mutant formed huge cell clusters and in SEM images the daughter cells were largely unseparated. To learn more about the function of the AmiA and GlcA, Bose et al.[10] investigated internal *amiA* or *glcA* deletions in *S. aureus*. The growth rate of the mutant was not impaired compared to the WT, however, both showed a clumping phenotype and both enzymes are essential for biofilm formation.

Atl forms a ring structure on the cell surface at the septal region for the next cell division site[24]. This peripheral ring consists of a large belt of PG in the division plane[25]. The resolution of this ring occurs within milliseconds ("popping"), and the separating cells split open asymmetrically leaving the daughters connected by a hinge[19,26]. By SEM, we observed, both in WT and the mutants, the peripheral ring as well as cracked

daughter cells with the typical features of a smooth and flat septal wall and a rough outer wall (Fig. 2). This might suggest that cell division is not principally affected in the Δ*amiA* and the Δ*glcA* mutants. However, when we examined dividing cells with TEM, we see significantly more differences between the Δ*amiA* and the Δ*glcA* mutants. The main difference is that, in 25–35% of the Δ*glcA* mutant cells, the division is asymmetric with a non-separated, kidney-shaped cell cluster; the asymmetric septum formation is indicated by red arrows (Fig. 4 and Supplementary Figs. 1 and 2). The dark spots seen in Δ*amiA* (white arrow) represent artifacts during sample preparation and imaging. In the Δ*glcA* mutant, the crosspeptides are hydrolyzed by the AmiA to generate naked glycan strands. The question is how (if) these unprocessed glycan strands cause this irregular septum formation. Wheeler et al. showed that particularly the lack of glucosaminidases Atl, SagA, ScaH, and SagB cause an increased surface stiffness and increased glycan chain length[27,28]. We assume that the asymmetric septum formation in the Δ*glcA* mutant is caused by the long unprocessed glycan strands, causing an increase in surface stiffness and a decrease in cell elasticity. Long unprocessed glycan strands and the resulting increased stiffness could affect the orthogonal septa formation for the next cell division thus causing the asymmetrical cell division. Digestion and processing of the septum can progress since AmiA does not need the presence and activity of GlcA. This result shows that GlcA is crucial for the generation of alternating perpendicular planes and thus for the symmetry of cell division. *S. aureus* cells divide into three alternating perpendicular planes, with sister cells remaining attached to each other after division[20]. With *glcA*, we have identified a gene/enzyme which contributes to a symmetrical cell division.

In the Δatl mutant, some part of the PG will be resolved because there are several minor amidases like the Aaa that are upregulated[4,8,9]. Moreover, in the Δatl mutant, a number of secondary PG hydrolases (Aaa, SsaA, Aly, SA2437, SA2097, SA0620, SA2353, SA2332, SA0710, SA2100, LytH, IsaA, LytM, SceD) were increased in the secretome and the corresponding genes were transcriptionally upregulated, suggesting a compensatory mechanism for the atl mutation[29]. Such back-up PG hydrolases might rescue the deleterious effect in cell separation in Δatl.

The structure of AmiA is resolved[15]. Its activity is described as a N-acetylmuramyl-L-alanine amidase, which cleaves the bond between the lactyl group of MurNAc and the peptide subunit of PG in several previous works. Oshida et al. and others described the activity of the amidase with zymogram analysis using heat-inactivated staphylococcal whole cells as amidase substrates and Micrococcus luteus as glucosaminidase substrates without showing any structural evidence[4–6]. Later Lutzner et al. developed a novel fluorescent substrate for S. epidermidis Atl amidase AmiE. But this substrate has no similarity with the natural PG since it neither has glycan components nor is the peptide sequence the same as the natural substrate[30]. Biswas et al. have used PG to show the mass difference extracted from an HPLC-MS analysis after a lysostaphin and amidase double digestion[23]. But surprisingly, none of these and other studies performed so far, have used the natural substrate, purified staphylococcal whole PG, and shown the specific activity of the Atl amidase. As there are enzymes with diverse hydrolytic activities, for example, LytN shows both amidase and peptidase activities when tested on staphylococcus PG[31], it is necessary and important to verify the activity spectrum of Atl amidase on its natural substrate. Furthermore, the crystal structure of the catalytic domain of AmiA in complex with a PG-derived ligand shows that the pentaglycine bridge is important for fitting into the binding cleft of AmiA[15]. The binding groove of the AmiA structure also explains why only the Lys-type PG of S. aureus can be hydrolyzed and not the meso-diaminopimelic acid (meso-DAP) type PG of B. subtilis. By treatment of PG with AmiA, more than 12 peaks could be resolved that correlated perfectly with the mass of mono- to >12-mer crosspeptides (Fig. 8a). This result shows the activity of AmiA with its natural substrate the staphylococcal PG. We also can demonstrate that it does not have additional PG-peptidase activity as seen for example with the amidase LytN[31]. The number of resolved peaks suggests that the glycan strands can be longer than 12 disaccharide units, with a predominant length of three and ten disaccharides. These results correlate well with earlier studies by Boneca et al.[32].

One of the most important part of this study was the finding that the glucosaminidase GlcA is not an endo- but an exo-β-N-acetylglucosaminidase. How it came about to denote the S. aureus glucosaminidase as an endo-β-N-acetylglucosaminidase is in retrospect not completely comprehensible[22,33]. Later, the characterization of the 51 kDa GlcA from S. aureus as an endo-beta-N-acetylglucosaminidase lacks clear experimental evidence as well[7], indicating that, in none of these reports an endo-activity has been really demonstrated. The crystal structures of S. aureus N-acetylglucosaminidase of the major autolysin (Atl) alone and in complex with fragments of PG revealed that N-acetylglucosaminidase of S. aureus and the muramidase (lysozyme) approach the substrate at alternate glycosidic bond positions[34]. The catalytic analysis revealed that reduced (MurNAc-GlcNAc)$_2$ as a substrate was cleaved to the disaccharides MurNAc-GlucNAc and MurNAc-GlcNAc$^{red}$, indicating that the staphylococcal GlcA is an N-acetylglucosaminidase; whereas the products MurNAc$^{red}$ and GlcNAc-MurNAc-GlcNAc$^{red}$ would have indicated a muramidase activity[34]. But, by using (MurNAc-GlcNAc)$_2$ as a substrate, it is not possible to distinguish between an endo- or exo-GlcA. It is difficult to distinguish if the disaccharide products are MurNAc-GlcNAc, GlcNAc-MurNAc, or a mixture of both. We think that the reducing end of the glycan strand is MurNAc since Lipid II is incorporated as GlcNAc-MurNAc[35].

GlcA cannot hydrolyze cross-linked PG. While AmiA digests PG to the variably long crosspeptides (Figs. 7a and 8a), GlcA did not produce any digestion products (Fig. 8b), indicating that cross-linked PG cannot be cleaved by GlcA. This explains why in zymogram the 52 kDa GlcA showed almost no bacteriolytic activity with S. aureus but with Micrococcus luteus cells[4]. This discrepancy was explained by the fact that M. luteus PG is less cross-linked than that of S. aureus[18]. In order to verify if cross-linked PG is not a substrate for GlcA, we double digested PG with AmiA and GlcA (Fig. 8c). While the amount of crosspeptides remains unaltered, two new predominant peaks appeared which turned out to be O-acylated and none-O-acylated MurNAc-GlcNAc disaccharides (Fig. 9a–c). This result confirms that GlcA is only active if the cross-linked peptides were previously removed by AmiA, confirming that the naked glycan strand is the substrate for GlcA. Similarly, the fact that we see only the two disaccharide peaks reveals that GlcA is an exo-N-acetylglucosaminidase. Had it been an endoenzyme, one would have expected intermediary oligosaccharides of different sizes. But this was observed at no time point. We only saw increasing peak areas for non-O-acylated (≈80%) and O-acylated (≈20%) MurNAc-GlcNAc peaks (Fig. 9d). PG O-acetylation is not only found in S. aureus but also in other pathogenic staphylococcal species Bera et al.[36] and contributes to increased pathogenicity by immune evasion[37]. Our results reveal that the GlcA activity is not hindered by O-acetylation of the glycan backbone in contrast to the muramidase lysozyme, as previously described[38].

Here, we unraveled three major differences between the muramidase lysozyme and GlcA. (i) Lysozyme hydrolyzes cross-linked PG, (ii) it represents an endoenzyme, and (iii) its binding to PG is markedly decreased by PG O-acetylation. On the other hand, GlcA cannot cleave cross-linked PG, rather it affords a 'naked' glycan strand; it is an exoenzyme and its activity is not inhibited by O-acetylation of MurNAc.

With this study, we show PG processing by Atl occurs in a defined order as illustrated in the model (Fig. 10). At first, the N-

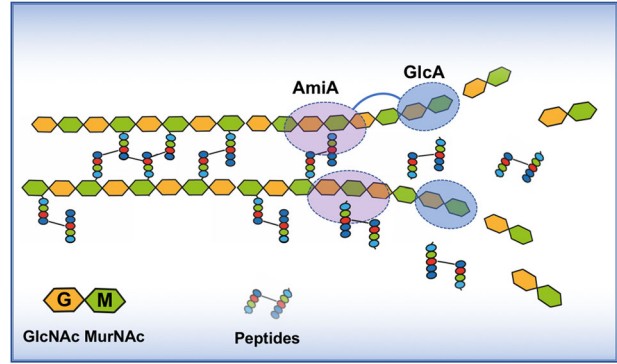

**Fig. 10 Model of staphylococcal PG processing by Atl.** PG processing by Atl occurs in a defined order. Upper strand: in the unprocessed Atl, AmiA and GlcA are linked together. AmiA slides along one strand of the PG and cuts off the crosspeptides exposing an unmodified "naked" glycan strand which is now substrate for GlcA. The exo-β-N-acetylglucosaminidase GlcA chops down the glycan strand into GlcNAc-MurNAc disaccharide units. Lower strand: proteolytically processed individual AmiA and GlcA act separately and are most likely involved in processing loose ends thus contributing to "smoothen" the CW.

acetylmuramyl-L-alanine amidase (AmiA) comes into play by hydrolyzing the crosspeptides leaving a naked glycan strand behind. This glycan strand is chopped down to GlcNAc-MurNAc disaccharides by the exo-β-N-acetylglucosaminidase (GlcA) following AmiA. In the unprocessed Atl, the two enzymes are combined in one protein in exactly the right catalytic order, AmiA-GlcA. This makes sense as AmiA must first free PG from crosspeptides before GlcA can be active. In both *S. aureus* and *S. epidermidis*, Atl is proteolytically processed to individual AmiA and GlcA enzymes. This also makes sense in so far as during the PG processing loose ends will be created that can be chopped down more easily by the individual enzymes thus contributing to "smoothen" the CW. Here, we have decoded the actual activity of glucosaminidase (GlcA) and show that the separation of the daughter cells takes place through a precisely coordinated interaction of AmiA and GlcA.

## Methods

**Bacterial strains, media, and growth conditions.** For all experiments *S. aureus* SA113, which is a NCTC 8325 derivative[39], its Δ*atl* deletion mutant SA113Δ*atl::spc* and the complemented mutant SA113Δ*atl::spc* (pRC14) were used[23,40]. Staphylococci were cultivated in tryptic soy broth (TSB), and *E. coli* DC10B in basic medium (BM) (1% soy peptone, 0.5% yeast extract, 0.5% NaCl, 0.1% glucose, and 0.1% $K_2HPO_4$, pH 7.2) at 37 °C and shaking at 150 rpm. When appropriate, the medium was supplemented with 10 μg/mL chloramphenicol for *Staphylococcus* strains and 100 μg/mL ampicillin for *E. coli* strains.

**Generation of markerless Δ*amiA* and Δ*glcA* deletion mutants in *S. aureus* SA113.** For the construction of the markerless *amiA* mutant, ~1000 bp fragment upstream of the catalytic domain of *amiA* and an ~1200 bp fragment downstream of its second repeat domain (R2) was amplified with primers Ami_Up Fwd/Rev and Ami_Down Fwd/Rev. Similarly, the markerless *glcA* mutant was generated by amplifying fragments of similar size upstream of the glucosaminidase repeat domain (R3) and downstream of its catalytic domain from SA113 genomic DNA using primers Glc_Up Fwd/Rev and Glc_Down Fwd/Rev (see supplementary Table 1 for a complete list of primers). Shuttle vector pBASE6[41] was linearized with EcoRV and the three fragments per construct were ligated by Gibson assembly[42]. Chemically competent *E. coli* DC10B cells, which lack dcm methylation and therefore allow direct plasmid transfer into *S. aureus*[43], were transformed with pBASE6-amiA_del and pBASE6-glcA_del plasmids, and successful transformants were selected and grown on LB agar plates supplemented with 100 μg/ml ampicillin. Electrocompetent SA113 cells were transformed by electroporation as described[44] with 2–4 μg of plasmid DNA (1 mm cuvette, 23 kV/cm), which was isolated and verified by the transformed *E. coli* DC10B cells. After electroporation, a prewarmed BM medium was added and the cells were further incubated for 2 h at 37 °C, which finally were plated on BM agar supplemented with 10 μg/ml chloramphenicol. The temperature-sensitive vector pBASE6 allows anhydrotetracycline-inducible expression of *secY* antisense RNA for counter-selection against the plasmid[41]. The deletion of both genes *amiA* and *glcA* was further conducted as described previously[41,45]. The correct sequence of the obtained plasmid constructs was verified by colony PCR and subsequent sequencing using the primers pBASE6_Fwd/Rev. Successful markerless deletion mutants were verified by PCR analysis and sequencing using primers AmiA_KO Fwd/Rev and GlcA_KO Fwd/Rev for *amiA* and *glcA* deletion, respectively.

**Cloning and expression of *S. aureus* amidase AmiA and glucosaminidase GlcA.** *amiA* with its two repeat regions *amiA-R1-R2* and *glcA* with its repeat domain *R3-glcA*[14,15] were cloned and expressed with an N-terminal His₆-tag in *E. coli* M15 using the IPTG inducible plasmid pQE30[23,46]. Cells were cultivated in 2xYT to $OD_{578nm}$ 0.5 at 37 °C, and then induced with 1 mM IPTG for 4 h at 20 °C. The cells were harvested by centrifugation and washed twice in PBS containing cOmplete protease inhibitor cocktail (Roche). The obtained cell pellets were then lysed using a French press. The crude extract was centrifuged (15 min, 5000 × *g* at 4 °C) and AmiA and GlcA enzymes were then purified using Ni-NTA superflow affinity chromatography (Qiagen) as described by the manufacturer. The imidazole elution fraction was further purified on a Superdex 75 size-exclusion chromatography column using 20 mM pH 7.5 sodium phosphate buffer containing 150 mM NaCl.

**Preparation of peptidoglycan (PG).** PG was isolated using the method developed by de Jonge et al.[21] with slight modification. Briefly, cells were grown until reaching an $OD_{578 nm}$ of 0.8 and harvested by centrifugation at 3000 × *g* for 30 min, washed twice with ice-cold 0.9% NaCl, boiled with 5% sodium dodecyl sulfate (SDS) for 30 min, washed and broken using 106 μM glass beads (Sigma) and FastPrep FP120 cell

milling apparatus (Bio 101, La Jolla, Calif.). Insoluble polymeric PG was harvested by centrifugation at 30,000 × *g* for 20 min and washed several times with lukewarm water to remove the SDS. Broken CWs were suspended in 100 mM Tris-HCl, pH 7.2, 20 μg/ml $MgCl_2$ and treated with 10 μg/ml DNase and 50 μg/ml RNase for 2 h at 37 °C. The broken cells were again centrifuged, resuspended with 2 M urea, and incubated for 1 h at room temperature. The broken cells were washed repeatedly with water to remove the urea and resuspended with 100 mM Tris/HCl, pH 7.2, and subsequently digested with 100 μg/ml trypsin for 16 h at 37 °C with the addition of 20 mM $CaCl_2$. To remove wall teichoic acid, the PG preparations were incubated on a magnetic stirrer with 48% hydrofluoric acid (HF) for 48 h at 4 °C. PG was harvested by centrifugation at 30,000 × *g* for 30 min and washed several times with water until the complete removal of HF. Washed PG was finally lyophilized and stored.

**HPLC analysis of muramidase, amidase, and glucosaminidase digested PG.** PG was analyzed basically as described previously[21] with some minor modifications. Purified PG was dissolved in 25 mM pH 6.8 sodium phosphate buffer and digested with the corresponding enzyme or enzyme combination for 16 h or the defined time. The resulting muropeptides or other corresponding products were reduced with sodium borohydride and separated by RP-HPLC using a Poroshell 120 EC-C18 (Agilent Technologies, Waldbronn, Germany) column (2.7-μm particle size, 150 × 4.6 mm, 120-Å pore size) and a linear gradient from 5% to 30% MeOH in 100 mM sodium phosphate buffer pH 2.2 at a flow rate of 0.6 ml/min. Peaks were detected at 205 nm.

**Mass spectrometric (MS) analysis and identification of AmiA and GlcA-digested PG fragments.** For the analysis of digested PG samples, an Acquity UPLC/Synapt G2 LC/MS system from Waters (Manchester) was used. A sample volume of 7 μl was injected onto a Waters Acquity C18 CSH 2.1 × 100 mm, 1.7-μm column, operated at a flow rate of 176 μl/min and a temperature of 52 °C. A 50 min gradient from 99% A (Water, 0.1% TFA) to 10% B (Methanol, 0.1% TFA) was used to separate the analytes, followed by a 48-min gradient from 10 to 15% B. The mass spectrometer was operated in ESI positive mode with a scan range from 50 to 2000 at a scan rate of 0.5 s/scan. Data were analyzed with Waters MassLynx software.

**Scanning (SEM) and transmission electron microscopy (TEM).** For SEM cells were fixed in 2.5% glutaraldehyde/4% formaldehyde in PBS for 2 h at room temperature and mounted on poly-L-lysine-coated coverslips. Cells were post-fixed with 1% osmium tetroxide for 45 min on ice. Subsequently, samples were dehydrated in a graded ethanol series followed by critical point drying (Polaron) with $CO_2$. Finally, the cells were sputter-coated with a 3-nm thick layer of platinum (CCU-010, Safematic) and examined with a field emission scanning electron microscope (Regulus 8230, Hitachi High Technologies) at an accelerating voltage of 5 kV. For TEM, cells were high-pressure frozen (HPF Compact 03, Engineering Office M. Wohlwend GmbH) in capillaries, freeze-substituted (AFS2, Leica Microsystems) with 2% $OsO_4$ and 0.4% uranyl acetate in acetone as substitution medium and embedded in Epon. Ultrathin sections were stained with uranyl acetate and lead citrate and analyzed with a Tecnai Spirit (Thermo Fisher Scientific) operated at 120 kV.

**Flow cytometric cell cluster analysis.** Cells were inoculated from an overnight culture to OD 0.1 and grown to mid-exponential phase. Samples were taken from each flask and diluted 1:100 in PBS and size distribution was analyzed using a forward scatter (FSC-A) measurement on a BD FACS Fortessa. Data were analyzed using FlowJo software.

**Reporting summary.** Further information on experimental design is available in the Nature Research Reporting Summary linked to this paper.

## Data availability

The authors declare that the main data supporting the findings of this study are available within the article and its Supplementary Information files. Extra data are available from the corresponding author upon request.

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

## Acknowledgements

This work was supported by the Deutsche Forschungsgemeinschaft (DFG) TRR 261, as well as by DFG, Germany´s Excellence Strategy—EXC 2124—390838134 "Controlling Microbes to Fight Infections". We acknowledge support by the Open Access Publishing Fund of the University of Tübingen.

## Author contributions

F.G. and M.N. designed the study. M.N., P.M.T., and F.G. designed the experiments. M.N. performed most of the experiments. M.S. performed MS analysis. K.H. performed EM analysis. P.M.T. contributed to manuscript writing and proofreading. F.G. and M.N. wrote the paper.

## Funding

## Competing interests

The authors declare no competing interests.
