## [Peer Review File · Communications Biology]

Reviewers' comments:

Reviewer #1 (Remarks to the Author):

The manuscript "New insights into the function and activity of the amidase- and glucosaminidase domain of the major autolysin (Atl) of *Staphylococcus aureus*" looks to differentiate the activities of the 2 domains of Atl and subsequently provide a model for Atl activity. Using Electron Microscopy & HPLC/Mass Spec the authors have shown that Atl is required for the splitting of daughter cells through sequential AtlA, GlcA activity. Deletions of these domains result in cells that are unable to properly split and have a rougher cell wall. Additionally, they showed that GlcA acts as an exo-glucosaminidase rather than an endo-glucosaminidase as previously suggested.

Overall the manuscript shows some good work and the methodology & conclusions are sound. There are a few instances where there are leaps made in the conclusions which could benefit from further evidence.

- The conclusions made from the EM data could do with additional quantification either from the images or with additional evidence: a) Quantify the "clumping" of the different strains e.g. through forward-scatter or another method. The Δ atl strain has previously been characterised as having increased clumping using forward scatter (Wheeler et al. 2015, Bose et al. 2012). b) Authors notice "asymmetric septa" for Δ glcA but not the other strains. Quantification of the % population across the 4 strains used would be preferential.

-The "thread-like" structures that are released in the mutants are presumed to be cell wall residues. Could these be varified e.g. through quantitative mass spec of culture supernatants to determine if any residues are enriched?

-It is my understanding that the autolysin ring (Yamada et al. 1996) and the 'piecrust' (Turner et al. 2010) are not the same things (although have the same localisation). The autolysin ring is on the outside of the cell while the 'piecrust' is a permanent architectural feature on the inner face of the PG. This is not clear in the author's discussion. Additionally, it is suggested these features "resolve", which is not the case. It is also not possible to visualise the 'piecrust' via SEM, only the interface between the rough outer wall and smooth septal surface.

-The authors suggest that GlcA plays a key role in symmetrical cell division, yet the full Atl deletion strain does not have increased asymmetric division. How do they reconcile this observation?

-It is concluded that the Atl domains (AtlA & GlcA) are responsible for cell splitting and the "smoothing" of the cell wall - however, this could easily be due to downstream effects, and therefore different proteins are key to this process. The authors could add purified enzymes to fixed/dead mutant cells to determine if the action of these proteins allows for the smoothing of the external CW.

-L 378-379, "We see a similar effect in the Δ glcA mutant but its defect cannot be rescued by SagA, ScaH and SagB". There is no evidence of this in the manuscript - where has this come from?

-Could determining the rate of hydrolysis of GlcA on WT and O-acetyl lacking PG show a lack of preference of GlcA for the O-acetylation state of substrate PG?

-For the HPLC experiments "equal amounts" of PG are used. It would be helpful to know by what measure this "equal" is. E.g. by weight or by cell c.f.u.

-Some figure legends are (direct) repeats/copies of the main text or contain information more suited to the main text rather than that of figure legends.

-Fig. 5. It is not clear where the staphylococcal PG originates from (Sa113 WT?)

-In the text (line 166) and figure legend 2, Reference incorrectly attributed (Zhu et al vs. Zhou et al.).

Reviewer #2 (Remarks to the Author):

This manuscript by Nega et al. reports on the in-depth characterization of the enzymatic activities represented by the amidase AmiA and the glucosaminidase GlcA domain of the major Staphylococcus aureus autolysin Atl. The authors used amiA, glcA, and atl mutants to define morphological changes, such as cell cluster formation and asymmetric cell division by SEM and TEM. Studies on the enzymatic activities of the peptidoglycan hydrolases using natural peptidoglycan prepared from S. aureus revealed that GlcA is only active upon removal of the stem peptide carried out by AmiA. Moreover, the observation that GlcA is an exo-enzyme removing disaccharides from the glycan backbone rather than an endo- β -N-acetylglucosaminidase as reported earlier was a rather unexpected finding. This is a very interesting manuscript that adds important new information to an extensively studied protein. The experiments were performed appropriately and the manuscript is technically sound and well written. However, there are some minor points of criticism.

1. In 102 / In 119 / Fig. 1A: These two paragraphs in the introduction section could be combined. The authors refer to the repeats R1ab/R2ab/R3ab and point out the presence of flexible linkers L1 and L2. The ab domains and linkers are missing from the Fig. 1A and should be added to the illustration for clarity reasons.
2. Abbreviations should be introduced at first appearance and then used throughout the manuscript. This should be checked, as it did not occur in each case, f.e. PG, CW, RP-HPLC.
3. In 166: There is no reference 107.
4. In 289: M. should mean Micrococcus
5. In 290: The references 4-6 do not all refer to Oshida et al.
6. In 321: "The crystal structures of S. aureus N-acetylglucosaminidase autolysin E (AtlE) alone and in complex with fragments of PG"... It is not clear what is meant with this part of the sentence as AtlE is the major autolysin from S. epidermidis.
7. In 351: There is no reference 116.
8. The manuscript contains some typing errors, i.e. In 308 (should be "demonstrate") and In 314 (should be "glucosaminidase"), which should be corrected.

Reviewer #3 (Remarks to the Author):

The manuscript by Nega et al. comes seeks to understand the contribution of Atl and its processed forms of AmiA and GlcA in S. aureus cell division and their activity. Of note, they identify for the first time that GlcA is not an endo enzyme as previously thought, but an exo enzyme. Overall, the paper is very well written with excellent figures, though perhaps Fig S2 should be separated. The data presented read largely observational, but could be slightly adjusted in writing to appear more hypothesis driven. The results are impactful to the Staphylococcus field and those interested in murein hydrolases.

I have only one major concern that I believe requires an experiment, the lack of a key control in Figure 7 (see below).

Line 91-94, this is known for S. aureus and should be referenced

Line 204, this is not surprising since amiA mutant more resemble atl mutants than glcA mutants. This should be stated at end. Or, I think the paper would sound less observational and more

hypothesis-driven if this was started with such as statement.

Figs 2-3. In figure 2, glcA and atl have rough surface, while amiA is intermediate and WT is relatively smooth. Yet, in Figure 3, amiA and glcA seem to show opposite results as is noted also in the figure legend. Can this be explained.

Digestion of PG....varying length. This is not surprising based on previous identification of AmiA cleavage. This should be noted and referenced.

Figure 7. While I believe the data in figure 7 to be true, it is missing an essential control which would be the heat-treated AmiA sample without GlcA.

There is no indication of how many times experiments were conducted or how many individual samples were analyzed. One has to only assume that the data are "representative". While I believe that the data in Fig 8D is no doubt significant, there is no statistical analysis. Is this because it was a single sample?

Discussion paragraph starting on line 373. The glcA mutant in Figure 3 does not resemble the atl mutant, despite the atl mutant lacking GlcA. This would indicate that AmiA is the cause of this division issue. Since AmiA works before GlcA, this is not completely addressed in this section and I would suggest expanding this paragraph to more clearly address this.

Referee expertise:

Referee #1: *S. aureus*, cell wall, cell division, peptidoglycan

Referee #2: autolysin, *S. aureus*

Referee #3: *S. aureus*, atl hydrolase

Reviewers' comments:

Reviewer #1 (Remarks to the Author):

The manuscript “New insights into the function and activity of the amidase- and glucosaminidase domain of the major autolysin (Atl) of *Staphylococcus aureus*” looks to differentiate the activities of the 2 domains of Atl and subsequently provide a model for Atl activity. Using Electron Microscopy & HPLC/Mass Spec the authors have shown that Atl is required for the splitting of daughter cells through sequential AtlA, GlcA activity. Deletions of these domains result in cells that are unable to properly split and have a rougher cell wall. Additionally, they showed that GlcA acts as an exo-glucosaminidase rather than an endo-glucosaminidase as previously suggested.

Overall the manuscript shows some good work and the methodology & conclusions are sound. There are a few instances where there are leaps made in the conclusions which could benefit from further evidence.

- The conclusions made from the EM data could do with additional quantification either from the images or with additional evidence:

- A) Quantify the “clumping” of the different strains e.g. through forward-scatter or another method. The Δ atl strain has previously been characterized as having increased clumping using forward scatter (Wheeler et al. 2015, Bose et al. 2012).
- B) b) Authors notice “asymmetric septa” for Δ glcA but not the other strains. Quantification of the % population across the 4 strains used would be preferential.

- We thank the reviewer for these suggestions. Using forward scatter we measured cell size distribution of the WT and mutants as suggested by the reviewer and shown in Fig. 3. We were able to confirm the observations we already made with microscopic analyses.

We wrote in lines 167 ff

This difference in cell cluster formation in the mutants was further confirmed by forward scatter flow cytometric measurements of WT SA113, Δ amiA, Δ glcA and Δ atlA cells grown to mid-exponential phase. The results clearly show that the mean cluster size increases in the order $WT < \Delta$ amiA < Δ glcA < Δ atlA confirming the previous microscopic observations (**Fig. 3**).

- We have quantified the proportion of cells with asymmetric septa across the 4 strains and shown it on supplementary Fig. 2.

We wrote in lines 279 – 282

The main difference is that, in 25-35% of the $\Delta glcA$ mutant cells, the division is asymmetric with a non-separated, kidney-shaped cell cluster; the asymmetric septum formation is indicated by red arrows (**Fig. 5, Supplementary Fig. S1, S2**).

-The “thread-like” structures that are released in the mutants are presumed to be cell wall residues. Could these be verified e.g. through quantitative mass spec of culture supernatants to determine if any residues are enriched?

We verified the thread like structures to be cell wall residues by adding purified enzymes to the aggregated Δatl cells as thankfully suggested by the reviewer and shown the results in Fig. 4.

We wrote in lines 173 ff

In order to confirm our hypothesis that the thread like structures are filaments of unprocessed ‘old’ PG, we digested heat inactivated mutant $\Delta atlA$ cells that were grown to mid-exponential phase with purified AmiA and GlcA. Comparison of the digested and undigested samples with SEM analysis showed that the clumping phenotype of the undigested sample is completely abolished and cells were separated. More importantly, the rough surface become smooth confirming that the ‘old’ PG was processed by the externally added enzymes (**Fig. 4**).

-It is my understanding that the autolysin ring (Yamada et al. 1996) and the ‘piecrust’ (Turner et al. 2010) are not the same things (although have the same localisation). The autolysin ring is on the outside of the cell while the ‘piecrust’ is a permanent architectural feature on the inner face of the PG. This is not clear in the author’s discussion. Additionally, it is suggested these features “resolve”, which is not the case. It is also not possible to visualise the ‘piecrust’ via SEM, only the interface between the rough outer wall and smooth septal surface.

The referee is right. To avoid confusion, we deleted the term 'piecrust'.

-The authors suggest that GlcA plays a key role in symmetrical cell division, yet the full *Atl* deletion strain does not have increased asymmetric division. How do they reconcile this observation?

Now we know that AmiA does not require GlcA for its activity. Therefore in the GlcA deletion mutant, AmiA is still active processing the septum formation even though the septation is not properly positioned. This is included in the discussion line 279-306:

The main difference is that, in 25-35% of the $\Delta glcA$ mutant cells, the division is asymmetric with a non-separated, kidney-shaped cell cluster; the asymmetric septum formation is indicated by red arrows (**Fig. 5, Supplementary Fig. S1, S2**). The dark spots seen in $\Delta amiA$ (white arrow) represent artifacts during sample preparation and imaging. In the $\Delta glcA$ mutant, the crossseptides are hydrolyzed by the AmiA to

generate naked glycan strands. The question is how (if) these unprocessed glycan strands cause this irregular septum formation. Wheeler et al. 2015 showed that particularly the lack of glucosaminidases Atl, SagA, ScaH, and SagB cause an increased surface stiffness and increased glycan chain length^{27,28}. We assume that the asymmetric septum formation in the $\Delta glcA$ mutant is caused by the long unprocessed glycan strands, causing an increase in surface stiffness and a decrease in cell elasticity. Long unprocessed glycan strands and the resulting increased stiffness could affect the orthogonal septa formation for the next cell division thus causing the asymmetrical cell division. Digestion and processing of the septum can progress since AmiA does not need the presence and activity of GlcA. This result shows that GlcA is crucial for the generation of alternating perpendicular planes and thus for the symmetry of cell division. *S. aureus* cells divide in three alternating perpendicular planes, with sister cells remaining attached to each other after division²⁰. With *glcA*, we have identified a gene/enzyme which contributes to a symmetrical cell division.

In the Δatl mutant some part of the PG will be resolved because there are several minor amidases like the Aaa that are upregulated^{4,8,9}. Moreover, in the Δatl mutant, a number of secondary PG hydrolases (Aaa, SsaA, Aly, SA2437, SA2097, SA0620, SA2353, SA2332, SA0710, SA2100, LytH, IsaA, LytM, SceD) were increased in the secretome and the corresponding genes were transcriptionally up-regulated suggesting a compensatory mechanism for the *atl* mutation²⁹. Such back-up PG hydrolases might rescue the deleterious effect in cell separation in Δatl .

-It is concluded that the Atl domains (AtlA & GlcA) are responsible for cell splitting and the “smoothing” of the cell wall - however, this could easily be due to downstream effects, and therefore different proteins are key to this process. The authors could add purified enzymes to fixed/dead mutant cells to determine if the action of these proteins allows for the smoothing of the external CW.

We thank the reviewer for this suggestion and have made the experiments accordingly. The results confirm our previous conclusion and are included in:

Lines 166 ff and Fig. 3:

This difference in cell cluster formation in the mutants was further confirmed by forward scatter flow cytometric measurements of WT SA113, $\Delta amiA$, $\Delta glcA$ and $\Delta atlA$ cells grown to mid-exponential phase. The results clearly show that the mean cluster size increases in the order $WT < \Delta amiA < \Delta glcA < \Delta atlA$ confirming the previous microscopic observations (**Fig. 3**)

-L 378-379, “We see a similar effect in the $\Delta glcA$ mutant but its defect cannot be rescued by SagA, ScaH and SagB”. There is no evidence of this in the manuscript - where has this come from?

The whole paragraph is edited Line 279 ff

-Could determining the rate of hydrolysis of GlcA on WT and O-acetyl lacking PG show a lack of preference of GlcA for the O-acetylation state of substrate PG?

In all the experiments we have done, we have not observed any indication that there is any difference in the rate of hydrolysis between the MurNAc- GlcNAc and its O-

acetylated form. Structural studies made on GlcA confirm our observation (Mihelic *et al.* ref. 31).

-For the HPLC experiments “equal amounts” of PG are used. It would be helpful to know by what measure this “equal” is. E.g. by weight or by cell c.f.u.

We have corrected the quantification description accordingly.

We wrote on line 197 ff:

For the analysis of PG crosslinking, PG isolated from the SA113, its Δatl mutant and complemented mutant Δatl (pRC14) were resuspended with buffer, adjusted to OD 3.0 and digested with mutanolysin.

-Some figure legends are (direct) repeats/copies of the main text or contain information more suited to the main text rather than that of figure legends.

Figure legends are corrected accordingly.

-Fig. 5. It is not clear where the staphylococcal PG originates from (Sa113 WT?)
All staphylococcal PG is isolated from the SA113 WT unless it is specified otherwise as indicated in the material and methods (lines 401-402). We corrected the sources accordingly (line 704).

-In the text (line 166) and figure legend 2, Reference incorrectly attributed (Zhu *et al.* vs. Zhou *et al.*).

Reference attribution is corrected (line 160).

Reviewer #2 (Remarks to the Author):

This manuscript by Nega *et al.* reports on the in-depth characterization of the enzymatic activities represented by the amidase AmiA and the glucosaminidase GlcA domain of the major *Staphylococcus aureus* autolysin Atl. The authors used *amiA*, *glcA*, and *atl* mutants to define morphological changes, such as cell cluster formation and asymmetric cell division by SEM and TEM. Studies on the enzymatic activities of the peptidoglycan hydrolases using natural peptidoglycan prepared from *S. aureus* revealed that GlcA is only active upon removal of the stem peptide carried out by AmiA. Moreover, the observation that GlcA is an exo-enzyme removing disaccharides from the glycan backbone rather than an endo- β -N-acetylglucosaminidase as reported earlier was a rather unexpected finding. This is a very interesting manuscript that adds important new information to an extensively studied protein. The experiments were performed appropriately and the manuscript is technically sound and well written.

However, there are some minor points of criticism.

1. In 102 / In 119 / Fig. 1A: These two paragraphs in the introduction section could be combined. The authors refer to the repeats R1ab/R2ab/R3ab and point out the presence of flexible linkers L1 and L2. The ab domains and linkers are missing from the Fig. 1A and should be added to the illustration for clarity reasons.

We combined both paragraphs.

2. Abbreviations should be introduced at first appearance and then used throughout the manuscript. This should be checked, as it did not occur in each case, f.e. PG, CW, RP-HPLC.

Abbreviations corrected.

3. In 166: There is no reference 107.

Typo error is corrected.

4. In 289: M. should mean Micrococcus

M. luteus is corrected to *Micrococcus luteus* (line 312).

5. In 290: The references 4-6 do not all refer to Oshida et al.

Reference citation is corrected to Oshida *et al.* and others (line 310).

6. In 321: "The crystal structures of *S. aureus* N-acetylglucosaminidase autolysin E (AtlE) alone and in complex with fragments of PG"... It is not clear what is meant with this part of the sentence as AtlE is the major autolysin from *S. epidermidis*.

The sentence was corrected as (Lines 343-346): The crystal structures of *S. aureus* N-acetylglucosaminidase of the major autolysin (Atl) alone and in complex with fragments of PG revealed that N-acetylglucosaminidase of *S. aureus* and the muramidase (lysozyme) approach the substrate at alternate glycosidic bond positions

7. In 351: There is no reference 116.

Typo error is corrected.

8. The manuscript contains some typing errors, i.e. In 308 (should be "demonstrate") and In 314 (should be "glucosaminidase"), which should be corrected.

Typo errors are corrected.

Reviewer #3 (Remarks to the Author):

The manuscript by Nega et. al. comes seeks to understand the contribution of Atl and its processed forms of AmiA and GlcA in *S. aureus* cell division and their activity. Of note, they identify for the first time that GlcA is not and endo enzyme as previously thought, but an exo enzyme. Overall, the paper is very well written with excellent figures, though perhaps Fig S2 should be separated. They data presented read largely observational, but could be slightly adjusted in writing to appear more hypothesis driven. The results are impactful to the Staphylococcus field and those interested in murein hydrolases.

I have only one major concern that I believe requires an experiment, the lack of a key control in Figure 7 (see below).

We appreciate this comment. We probably didn't make clear what the HPLC profiles mean. In Fig. 9A we have digested purified PG with AmiA. After 16 h digestion, the reaction was stopped by treating the sample at 95°C for 3 min to inactivate AmiA. From this sample we took an aliquot and digested it further with GlcA (Fig. 9C).

Therefore, the sample shown in Fig. 9A is our control. The figure legend is corrected accordingly.

Line 729: All samples are heat-inactivated at 95°C for 3 min before analysis

Line 91-94, this is known for *S. aureus* and should be referenced
The whole paragraph is referenced properly.

Line 204, this is not surprising since *amiA* mutant more resemble *atl* mutants than *glcA* mutants. This should be stated at end. Or, I think the paper would sounds less observational and more hypothesis-driven if this was started with such as statement.

We have changed the discussion accordingly. Lines 279 f

The main difference is that, in 25-35% of the $\Delta glcA$ mutant cells, the division is asymmetric with a non-separated, kidney-shaped cell cluster; the asymmetric septum formation is indicated by red arrows (**Fig. 5, Supplementary Fig. S1, S2**). The dark spots seen in $\Delta amiA$ (white arrow) represent artifacts during sample preparation and imaging. In the $\Delta glcA$ mutant, the crossseptides are hydrolyzed by the *AmiA* to generate naked glycan strands. The question is how (if) these unprocessed glycan strands cause this irregular septum formation. Wheeler *et al.* 2015 showed that particularly the lack of glucosaminidases *Atl*, *SagA*, *ScaH*, and *SagB* cause an increased surface stiffness and increased glycan chain length^{27,28}. We assume that the asymmetric septum formation in the $\Delta glcA$ mutant is caused by the long unprocessed glycan strands, causing an increase in surface stiffness and a decrease in cell elasticity. Long unprocessed glycan strands and the resulting increased stiffness could affect the orthogonal septa formation for the next cell division thus causing the asymmetrical cell division. Digestion and processing of the septum can progress since *AmiA* does not need the presence and activity of *GlcA*. This result shows that *GlcA* is crucial for the generation of alternating perpendicular planes and thus for the symmetry of cell division. *S. aureus* cells divide in three alternating perpendicular planes, with sister cells remaining attached to each other after division²⁰. With *glcA*, we have identified a gene/enzyme which contributes to a symmetrical cell division.

In the Δatl mutant some part of the PG will be resolved because there are several minor amidases like the *Aaa* that are upregulated^{4,8,9}. Moreover, in the Δatl mutant, a number of secondary PG hydrolases (*Aaa*, *SsaA*, *Aly*, SA2437, SA2097, SA0620, SA2353, SA2332, SA0710, SA2100, *LytH*, *IsaA*, *LytM*, *SceD*) were increased in the secretome and the corresponding genes were transcriptionally up-regulated suggesting a compensatory mechanism for the *atl* mutation²⁹. Such back-up PG hydrolases might rescue the deleterious effect in cell separation in Δatl .

Figs 2-3. In figure 2, *glcA* and *atl* have rough surface, while *amiA* is intermediate and WT is relatively smooth. Yet, in Figure 3, *amiA* and *glcA* seem to show opposite results as is noted also in the figure legend. Can this be explained.

We think that these differences arise not from the difference in surface structures or enzyme activities, but rather in sample preparation and imaging procedures of the TEM. We have corrected the description in the paragraph.

We wrote in the lines 282 -283: The dark spots seen in $\Delta amiA$ (white arrow) represent artifacts during sample preparation and imaging.

Digestion of PG.....varying length. This is not surprising based on previous identification of AmiA cleavage. This should be noted and referenced.

Previous works are described and discussed extensively and references are made accordingly in the paragraph line 308 ff.

Figure 7. While I believe the data in figure 7 to be true, it is missing an essential control which would be the heat-treated AmiA sample without GlcA.

We appreciate this comment. We probably didn't make clear what the HPLC profiles mean. In Fig. 9A we have digested purified PG with AmiA. After 16 h digestion, the reaction was stopped by treating the sample at 95°C for 3 min to inactivate AmiA. From this sample we took an aliquot and digested it further with GlcA (Fig. 9C). Therefore, the sample shown in Fig. 9A is our control. The figure legend is corrected accordingly.

Line 721: All samples are heat-inactivated at 95°C for 3 min before analysis

There is no indication of how many times experiments were conducted or how many individual samples were analyzed. One has to only assume that the data are "representative". While I believe that the data in Fig 8D is no doubt significant, there is no statistical analysis. Is this because it was a single sample?

Text added to the figure legend line 729-730: Figure is representative of three independent experiments.

Discussion paragraph starting on line 373. The glcA mutant in Figure 3 does not resemble the atl mutant, despite the atl mutant lacking GlcA. This would indicate that AmiA is the cause of this division issue. Since AmiA works before GlcA, this is not completely addressed in this section and I would suggest expanding this paragraph to more clearly address this.

We have edited the discussion accordingly. Lines 279 ff

REVIEWERS' COMMENTS:

Reviewer #1 (Remarks to the Author):

The manuscript of "New insights into the function and activity of the amidase- and glucosaminidase domain of the major autolysin (Atl) of *Staphylococcus aureus*" has been changed in response to my previous review. I feel that the new experiments and analysis cover my previous comments and back up the claims made previously with sufficient evidence.

I am happy to approve this version of the manuscript and feel it will be of interest to those in the cell division field.

Reviewer #3 (Remarks to the Author):

I feel like the authors have adequately addressed the concerns of myself and the other reviewers. I have no further suggestions.